# Polly Wants a Genome: The Lack of Genetic Testing for Pet Parrot Species

**DOI:** 10.3390/genes12071097

**Published:** 2021-07-20

**Authors:** Henriëtte van der Zwan, Rencia van der Sluis

**Affiliations:** Focus Area for Human Metabolomics, North-West University, Potchefstroom 2531, South Africa; Rencia.Appelgryn@nwu.ac.za

**Keywords:** psittaciform, plumage colour variation, molecular tools

## Abstract

Parrots are considered the third most popular pet species, after dogs and cats, in the United States of America. Popular birds include budgerigars, lovebirds and cockatiels and are known for their plumage and vocal learning abilities. Plumage colour variation remains the main driving force behind breeder selection. Despite the birds’ popularity, only two molecular genetic tests—bird sexing and pathogen screening—are commercially available to breeders. For a limited number of species, parentage verification tests are available, but are mainly used in conservation and not for breeding purposes. No plumage colour genotyping test is available for any of the species. Due to the fact that there isn’t any commercial plumage genotype screening or parentage verification tests available, breeders mate close relatives to ensure recessive colour alleles are passed to the next generation. This, in turn, leads to inbreeding depression and decreased fertility, lower hatchability and smaller clutch sizes, all important traits in commercial breeding systems. This review highlights the research carried out in the field of pet parrot genomics and points out the areas where future research can make a vital contribution to understanding how parrot breeding can be improved to breed healthy, genetically diverse birds.

## 1. Introduction

There are 356 extant psittaciform, or parrot, species natively found mainly around the tropical and sub-tropical areas of South-America, Africa, Asia and Australasia [1]. The most noticeable feature of parrots is their short, blunt bill with a down curved upper mandible over the upturned lower mandible [1]. In a study by [2], it was found that nearly one third of these species are threatened by extinction due to illegal parrot trade and many more are affected due to habitat destruction. The Psittaciformes group has the highest aggregate extinction list by the International Union for Conservation of Nature (IUCN Red List Index) compared to any other bird group [2]. With the exception of *Agapornis roseicollis* (peach-faced lovebird), *Melopsittacus undulatus* (budgerigar or budgie) and *Nymphicus hollandicus* (cockatiel) all species in the psittaciform group are protected by the Convention of International Trade in Endangered Species (CITES) [3,4]. In many countries worldwide, due to strict legislation, pet parrots are legally sold in four ways: (1) they are lovebirds, budgerigars and/or cockatiels, which are not protected under CITES; (2) they were imported from wild-caught parents prior to legislation being passed; (3) they are imported with special permits; or (4) they are born and raised in captivity. In a policy paper by the Animals & Society Institute in 2013 [5] it is clearly stated that the negative effects of confinement, i.e., physical ailments or traumatic experiences, compromise the well-being of parrots, and subsequently no parrot species is a viable candidate for captivity. They do not specify whether species such as budgerigars, lovebirds and cockatiels are viable candidates. Nevertheless, parrots are considered the third most popular pet (after dogs and cats) kept in the United States of America [3]. In 2017, it was estimated that there were over 49 million birds kept as pets in European households [6] and 2.6 million birds kept in Indonesian cities [7], and parrots continue to be the most popular bird species to keep in China [4].

Therefore, even though parrots are not necessarily considered “domesticated” species, the vast number of parrots kept as pets worldwide should motivate more research into genetic tools to aid breeders in breeding healthier and genetically diverse birds. There are many studies on parrot conservation in their natural habitat. A few examples of recent studies include a genetic study by [8] on the heterozygosity levels of the critically endangered *Neophema chrysogater* species; the effect of global habitat destruction on parrot species, especially in the Neotropics and Oceania areas [9]; and conservation efforts in Honduras to protect *Ara macao cyanoptera* [10]. Very little attention is currently focused on studies to improve the breeding of healthy, genetically diverse pet birds. With millions of birds kept as pets across the globe, all species of parrot (with the exception of lovebirds, budgerigars and cockatiels) cannot be continually viewed as “exotic pets”, and there has to be a shift towards breeding birds with the focus on genetic health and biodiversity. This has been successfully achieved in other pet breeding systems (e.g., dogs and cats) where the development of molecular genetic tools improved breeding systems (see next paragraph for examples where tests were developed and applied). These tools can be used, not only for breeders, but in programs where birds are especially bred to be reintroduced into the wild. This was demonstrated in the study by [8], where SNP data were used to assess the success of such a breeding program in the critically endangered *Neophema chrysogater* species.

Many molecular genetic tests are commercially available for dogs and cats such as individual identification, parentage verification panels [11,12] and health screening tests (e.g., progressive retinal atrophy in Golden Retrievers [13], or polycystic kidney disorder in cats, [14]). Similarly, coat colour genetic testing is also widely used (e.g., Siamese and Burmese patterns in cats, [15] and an array of coat colorations in dogs [16]). In the case of parrots, only a handful of individual and family identification panels have been developed (discussed in more detail later in this manuscript) and mainly for conservation use, not intended as a commercially available test for breeders. Surprisingly, thus far only one gene linked to plumage coloration has been identified (see Section 5 on plumage colour variations for more detail). As far as health screening goes, no genetically linked anomalies are routinely tested for in parrot species, as is the case in other pet species such as dogs and cats. The molecular screening tests most frequently used include sex determination (especially for birds where genders are similar in colour) [17] and pathogen testing to identify the beak and feather disease virus (BFDV) causing Psittacine beak and feather disease (PBFD) [18]. This review aims to highlight the progress made in terms of research for these species, point out the current gaps with regard to genetic testing in pet parrot species and discuss opportunities for research in this field. This review will focus on parrots as pets and not conservation of wild species.

## 2. Parrot Domestication

The mummified skeletons of 27 parrots from six species of macaw, amazons and conures found across five archaeological sites in the Atacama Desert in northern Chile, indicate that parrots were transported, captured and kept as pets and a symbol of wealth as early as 1100–1400 CE [19]. It is also documented that as early as 327 B.C., when Alexander the Great invaded India, one of the ancient biographers, Arrian, wrote of a parrot uttering a human voice [20]. Lovebirds have been kept as pets since the eighteenth century, with reports of *Agapornis pullarius* being bred in Britain in the 1880s [21]. Small parakeets such as lovebirds and budgerigars are popular in aviculture due to the fact that they breed relatively easy in captivity, are hardy, active and have a range of plumage colours [22].

Domestication of animals or birds is a broad term and a few definitions as given by [21]:1.Domestication is defined as that condition wherein the breeding, care and feeding of animals are more or less controlled by humans and/or;2.The adaptation to captivity via population genetic mechanisms in which natural selection is largely replaced by artificial selection.

The most recognized attribute of a “domesticated” vs. “wild” individual of the same species, is that domesticated individuals differ from their wild relatives in many ways, due to genetic and environmental factors [21]. This definition immediately brings colour variations of parrots to mind, but many of these variations also occur in natural populations [23]. However, reports of changes in wing-shape (e.g., [24], see Section 8 on health and behaviour screening tests for more details) in captive bred parrots might highlight the fact that some captive-bred species cannot be re-introduced after generations of breeding. This should also be taken into account in programs where birds are especially bred to be reintroduced to wild populations to prevent extinction. An alternative view on this topic is given by [25], stating that these adaptations due to artificial selection are not all necessarily bad, and could help a species survive in captivity rather than to go completely extinct. 

## 3. Genomic Research

The last decade saw the sequencing, assembly and annotation of 31 parrot genomes (see Table 1 for a list of some of the genomes). These reference genomes act as the blueprint from which SNPs, genes and other genetic markers of interest can be identified that could aid in the development of genetic tests for breeders and conservationists. Sequencing and assembling of genomes is not crucial to the development of molecular genetic tests and many markers can be identified without a genome. However, without a draft genome and since many de novo genomes are only assembled up to scaffold level, it complicates the comparison of data between different species. De novo genomes are essential for genotype-to-phenotype association studies, genome comparison studies especially in genome evolution and speciation studies [26]. It is therefore easier to develop cross-species molecular tools from chromosome-level assembled genomes.

The average size of the parrot genomes listed in Table 1 were 1.1–1.2 Gb, with the exception of the Puerto Rican parrot genome that was found to be 1.58 Gb. Between 14,000 and 16,000 genes were annotated for these genomes. In a recent study by [36], an additional 22 parrot genomes were sequenced. It is, however, unclear from that manuscript whether these genomes were annotated, and this article was not peer reviewed.

## 4. Selection Criteria Used in the Breeding of Parrots

One of the most striking features of most parrot species are their plumage coloration, and therefore it is not surprising that this is also the main economic trait they are selected for [37]. Parrots are also known for their cognitive abilities as well as vocal learning (see [38] for extensive research on the abilities of African grey parrots (*Psittacus erithacus*)). Factors influencing parrot online sales volume in China was investigated by [4] and it was found that generation length and body mass were the two determining factors that contributed to higher sale price. In Taiwan, native and not endangered bird species as well as birds without yellow coloration and without attractive songs were sold for less than other species [39]. 

In Indonesia, parrot keeping, in particular for lovebirds (*Agapornis* spp.) is a big industry [40]. In this country, singing contests are a major attraction where birds compete against each other and whichever bird sings the longest, wins. There are more than 10,000 such contests each year, in sharp contrast to the 20–30 traditional bird shows or auctions in Indonesia. It doesn’t seem that there is any genetic selection involved in these contests, and each contestant simply enters any bird they think will sing the longest [40]. Since Indonesia was identified as one of the countries with the highest priority for parrot conservation [2], contests such as these might drive illegal poaching and trade of these birds. 

The only selection criteria based on phenotypic Mendelian inheritance patterns in parrot breeding is plumage coloration. Birds with rare colour variations could sell for up to 700 times more than wildtype birds in Europe and up to 1300 times more in Asia [37]. In two of the most popular pet parrot groups, budgerigars and lovebirds, there are over 30 recognized inherited colour variations, most of which are inherited as autosomal recessive traits [37,41]. Since there is no genotyping test available to breeders, the only way to ensure the inheritance of a recessive allele is by mating close relatives. 

## 5. Plumage Colour Variations

Some colour variations found in parrots have been reported in wild birds as early as the first half of the twentieth century. In 1932, a blue *Agapornis personatus* male was reported in the Proceedings of the Zoological Society of London [42,43], and colour variations in wild lovebirds in 1942 and 1948, respectively, were observed by [44], whereas a lutino *A. roseicollis* was identified in a wild flock in Namibia [45]. Most of these variations are inherited as autosomal recessive traits, leading many breeders to mating close relatives in order to increase their chances to produce a bird with a rare plumage [37]. This could create major genetic diversity issues due to inbreeding depression and health issues (see [46,47] for examples of the effect of inbreeding depression). Research into the genetic control of colour traits has been extremely limited, which highlights the importance of further studies in this field. The European Association of Zoos and Aquaria, Parrot Taxon Advisory Group has compiled a list of parrot species where plumage variations are often observed [48] and can be found in Appendix A. Data in this table are based on observations and not published data.

Two pigment types are found in the plumage of parrots: black, brown and red-brown colours in the plumage, claws, beaks and eyes are caused by the melanin pigments, whereas green, blue, purple, red, yellow, pink and orange plumage colours are caused by psittacofulvin pigments [49,50]. Parrots are the only species on earth known to produce psittacofulvin pigments and the expression of these pigments is controlled genetically and not environmentally [49,51,52,53]. Five psittacofulvin pigments across 27 parrot genera were identified by [53]. Plumage colour is created by the combination of pigment and structural modifications in the barbs of parrot feathers [54]. Psittacofulvin pigment is in fact yellow, but due to light wave interactions with the ß-keratin rods, the light is reflected irregularly and appears green [55,56,57,58]. The genetic control of the psittacofulvin and ß-keratin systems is not yet fully understood, but these two systems are most likely under independent genetic control [50]. Most of the inheritance patterns of known colour variations found amongst lovebirds and budgerigars have been confirmed by test matings as autosomal recessive or sex-linked recessive traits [37,41].

There are four different mechanisms that cause colour variations in parrots. Many colour variations are caused by a change in melanin distribution [49,50] or a reduction in eumelanin production [37]. A change in the feather structure has also been shown to cause different plumage colours [55,56]. In the two studies by [55,56], it was shown that the spongy zone of the feathers in lovebirds with a darker green plumage (phenotype called “dark green”) is smaller than in wildtype birds, and the trait is inherited as an autosomal dominant trait with incomplete penetrance [37]. Similarly, in budgerigars, the slate phenotype is inherited as a sex-linked recessive trait, which causes the feather structure to be different between wildtype and slate birds [59]. The fourth mechanism involves the reduction or total abolishment of production of psittacofulvin [55,56]. The only mechanism where the gene and polymorphism responsible for a change in plumage colour has been identified, is due to a reduction in psittacofulvin pigment [55,56].

Green feathers contained yellow psittacofulvin pigment, whereas blue feathers contained no psittacofulvin pigment [55,56]. Through test matings, it was confirmed that blue feathers were inherited as an autosomal recessive trait [37]. A polymorphism causing green to blue colour inheritance in budgies was identified by [60]. This mutation was confirmed to be a T > C substitution mutation at position 1930 in the *MuPKS* gene, resulting in a single amino acid substitution (R644W). In this study, all 162 wildtype green birds had genotype R/R or R/W and all 118 blue birds had the W/W genotype. This SNP is located on Scaffold NW_004848279.1 on the budgerigar genome [29] at position 21369066. It still needs to be confirmed whether this mutation results in the same phenotype observed in lovebirds and other parrot species.

In the “blue-series” of parrot coloration, phenotypes called “aqua”, “turquoise” and “sapphire” are also found (see Figure 1 for some examples). There are no reports where the psittacofulvin has been quantified, but it is estimated that around 40–50% of the normal psittacofulvin pigment is produced in the feathers of aqua and turquoise birds [37]. These phenotypes, too, are inherited as autosomal recessive traits and further genetic studies are needed to show that the *MuPKS* gene might be linked to this inheritance pattern.

## 6. Parentage and Individual Identification Research

As mentioned above, the trait most widely used to select parrots as breeding stock is colour variation. Many of the popular parrot species kept as pets, e.g., lovebirds and budgies, have up to 30 different colour variations, most of which are inherited as Mendelian traits. Despite this, the genes and mutations linked to the variations are yet to be discovered [37,41]. Therefore, should a chick hatch with wildtype coloration, breeders use pedigree data (including the parents’ coloration) to predict the probability of that bird being a heterozygote of a specific variation. Pedigree information must therefore be accurate and complete, but this is not always the case. A routinely available commercial parentage verification test that a breeder or buyer of a bird can request before a sale is made, is the solution to incomplete and inaccurate pedigree information. The development of such test will, however, require extensive research into allele frequencies of populations, both breeding birds and wild populations, across different species and sub-species. 

Despite the wide use of parentage tests as a selection tool in the production of animals and poultry [61,62,63] as well as pet breeding [64,65], the application of parentage and individual identification tests in aviculture, and especially parrot breeding, has been very limited. To date, only four parentage verification panels have been developed for parrots, three of which were microsatellite-based and one SNP-based. These include a 16 microsatellite marker panel to verify parentage of the Cape parrot (*Poicephalus robustus*) [66]; 106 microsatellites amplified amongst seven different parrot species (*Amazona brasiliensis*, *A. oratrix*, *A. pretrei*, *A. rhodocorytha*, *Anodorhynchus leari*, *Ara rubrogenys* and *Primolius couloni*) and could be used for individual identification [67], and a panel of 11 microsatellite loci for African grey parrots [68]. All three of these studies were aimed at conservation and prevention of illegal trade of wild parrots, and not as a breeding tool. A SNP-based parentage verification panel for lovebirds which included 195 SNPs was developed especially for application in breeding systems [69]. This was the first study to develop a SNP-based parentage verification panel for any parrot species, and especially with the specific aim to be used as a selection tool. No parentage test currently exists for popular pet species such as budgies, Indian ring necks (*Psittacula krameri)* or any of the macaw species. 

The lack of commercially available parentage verification tests creates a major problem in genetic diversity, as close relatives are mated to ensure rare genetically inherited colour variations are passed down to the offspring. The effects of inbreeding depression were clearly demonstrated in two critically endangered bird species. The first was on the kakapo (*Strigops habroptilus*), a flightless parrot native to New Zealand [46]. They found that in a small breeding population of 51 individuals, females with lower heterozygosity had smaller clutch sizes and lower hatchability rates. The second was on the pink pigeon (*Columba mayeri*) where inbreeding depression was shown to negatively affect egg fertility, squab, juvenile and adult survival [47]. The effect was evident in free-living and captive bred birds. The development of a SNP chip for more parrot species that confirm individual and family identity could alleviate this problem.

## 7. Sexing

Determining the sex of a mating pair, or any bird for that matter, is important in any breeding system. Many parrot species are monomorphic and males and females cannot be distinguished externally [70]. The Chromo-Helicase DNA (*CHD*) binding site gene has long been used as a universal marker to identify or confirm sex in many bird species [71,72,73,74]. Female birds carry two types of sex chromosomes (W and Z), whereas males carry only one pair (Z). The W chromosome (as with the Y chromosome in mammals) has lost almost all of its genes and is much smaller in size [71,74]. The *CHD* gene encodes a protein that is involved in global regulation of transcriptional activation at chromatin level and exist in two genomic copies—one on the W chromosome (CHD-W) and one on the Z chromosome (CHD-Z) [70]—and both copies can be amplified with one set of primers. The resulting polymerase chain reaction (PCR) amplification products are two bands on a gel electrophoresis in females (W and Z) and one band in males (W) [72,73,74]. 

Many genetic service provider laboratories offer this test as a commercially available sexing option for parrot owners and breeders (e.g., Animal genetics laboratories (https://www.animalgenetics.us/Avian/DNA_Sexing/DNA-Sexing-Index.asp, accessed on 8 July 2021); BioBest laboratories (https://biobest.co.uk/avian-sexing/, accessed on 8 July 2021) and Lumegen laboratories (https://www.lumegen.co.za/avian-birds-home/, accessed on 8 July 2021)). The test is, however, labour intensive and prone to human error since a laboratory technician needs to interpret the results. 

A recent study developed a novel qPCR-based sexing method by comparing gene copy number variation in a selection of conserved Z-specific genes. Birds from both sexes, from 73 bird species covering 22 orders were tested at nine Z-specific loci (*CHRNA6*, *DDX4*, *DOCK8*, *FUT10*, *LPAR1*, *PIGG*, *PSD3*, *TMEM161B*, *VPS13A*) and three autosomal genes (*GGPS1*, *KIAA1429*, *MECOM*) [75]. These species included seven parrot species (*Amazona aestiva* (two males, two females), *A. amazonica* (one male, two females), *A. auropalliata* (two males, two females), *Ara ararauna* (one male, one female), budgerigar (two males, two females), *Pionites melanocephalus* (one male, one female) and *Psittacus erithacus* (one male, one female)). The Z-specific genes *CHRNA6*, *DDX4*, *THEM161B* and *VPS13A* were useful in determining the sex of neognath species (the bird clade which includes most bird species) and *DOCK8*, *FUT10*, *PIGG* and *PSD3* could identify the sex in paleognath birds (the clade which includes flightless species). In addition, the *LPAR1* gene could identify sex in both clades. It was suggested that the three autosomal genes could be used for normalization and autosomal controls in both clades. As found by Kroczak et al., [76] not all genes amplified in all species. 

Kroczak et al. [76] compared markers and polymorphisms that have been used routinely and commercially as “universal markers” to identify sex in birds and found that no single universal marker can be applied to sex all birds, especially parrots, successfully. These included the two widely used markers *P2P8* and *CHD1iA* as well as four W/Z length polymorphisms (*CHD1iE*, *CHD1i16*, *CHD1i9* and *NIPBLi16*). Their findings suggest that neither the two markers nor the four polymorphisms, when used alone as a single marker, were able to correctly determine the sex of all 135 parrot species. They found *CHD1iA*, *CHD1i0* and *NIPBLi16* to perform the most consistently across species. None of these markers could, when used as a single marker, successfully identify sexes of all species tested. They propose the combination of any two of the following four markers: *CHD1i16*, *CHD1i9*, *NIPBLi16* and *CHD1iA*, instead of only one universal marker.

## 8. Health and Behaviour Screening Tests

As mentioned earlier, there are no genetic health screening tests for inherited disorders or anomalies in parrots. Due to the level of inbreeding in some species, especially to establish recessive colour variations, these disorders are most probably inevitable at some point. In a rather old study by [77], two breeding programs were used to determine the impact of inbreeding on budgerigars. They found contradicting results, as in the one flock inbreeding had no effect on the clutch size and fledging success of the chicks, and in the other flock it either had no impact or it increased the clutch size, fertility and hatchability. They concluded that this could be as a result of tolerance of the genotype in the flock was a trait that increased, or that it could be as a modification as a result of domestication. They recommended further research, but no follow up studies could be found. This is in direct contrast to the findings of more recent studies on the Kea and Pink pigeon, where inbreeding had a direct negative effect on clutch size, hatchability, egg fertility, squab, juvenile and adult survival [46,47].

In a recent study [24] it was found that the wing shape of the critically endangered orange-bellied parrot (*Neophema chrysogaster*) (considered the world’s most endangered parrot), which has been bred in captivity since 1986, was significantly changed compared to wild birds of the same species. They found that the first two flight feathers were shorter in captive birds, and the fifth and sixth feathers were longer than those found in wild birds. This phenomenon might compromise a bird’s ability to survive in the wild, should it be re-introduced. Further studies on other species are lacking. 

Examples of negative behaviours displayed by many parrots in captivity are those of feather picking [78] and stereotypies (sequences of movements that are repeated identically, where such movement are inappropriate, lacing any function or goal) [79,80]. In a study of 64 Orange-winged amazon parrots (*Amazona amazonica*), it was established that feather picking has a heritability value (heritability is a measure of the strength of the relationship between performance (phenotypic values) and genotypic values for a trait in a population [81].) of 1.14 ± 0.27 [82]. Even though there are environmental factors that also influence this behaviour, the heritability value indicates that it could be possible to select against this behaviour. Two species, African grey parrots and cockatoos, were found to be more likely to display feather picking behaviour [83]. More research into this behaviour and the genetic basis thereof can provide a solution to a behaviour problem displayed in may parrot species. 

## 9. Future Research

There is an immense lack of research being carried out in the field of parrot breeding. There are many groups working on parrot conservation and reintroduction of rehabilitated parrots into the wild, but very few are focusing on molecular genetic research of pet parrots. When taking the popularity of parrots as pets around the world into consideration as well as the negative effect of inbreeding, the importance of future research on colour polymorphisms, sexing, species identification, health screening and quantitative trait selection cannot be ignored. As discussed earlier in this paper, identifying genes and polymorphisms linked to colour variations could reduce inbreeding, as breeders genotype birds at the specific colour loci before a mating takes place. Similarly, the inclusion of polymorphisms linked to sex could simplify the current sexing test available to breeders and SNPs linked to family relationships and identity in more parrot species must be identified. A combination of two to four of the polymorphisms tested by Kroczak et al. [76], could be included on the SNP chip. It should, however, be noted that the Z chromosome (shared between males and females) is euchromatic and gene-rich, while the W chromosome (only found in females) is gene-poor [84]. Xu and Zhou [84] found that despite the independent evolution of recombination suppression, the female W chromosome in birds still consists of highly conserved gene content. Further research on polymorphisms located on the W chromosome that can be combined with the polymorphisms suggested by [76] is needed before it can be included on a genus-specific SNP chip. Autosomal genes, additional to those used by [75], linked to male or female trait expression could be identified in addition to Z-linked markers.

Although a genus-specific SNP chip could benefit breeders and conservation officials in the fight against illegal poaching and trade, the cost of developing and validating this technology is still relatively high. High costs might make routine testing of birds by breeders impossible. 

Hybridization of species as well as genetic introgression is not uncommon in birds [85], especially in a closed colony breeding system where different species of the same genus (e.g., lovebirds) are kept (see Figure 2 for examples of a hybrid bird between two *Agapornis* species). Breeders also use the crossing of species to introduce colour variations found in one species to another species, or to create a “new” colour variation by combining two colours from two different species [41]. Hybridization could be viewed as either “good” or “bad”, depending on the application. In breeding this could be viewed as a “good” thing, since a new species or colour variations could be developed which could, in turn, boost sales. In conservation, this could have negative effects since heterogametic hybrid offspring are often less viable and less fertile than pure birds [85]. Most hybrids are not accepted in breeding societies and at auctions or shows, as only pure birds are registered at these bodies [41]. By identifying SNPs linked to specific species it might allow registration bodies to confirm the species of a bird at a show or auction. Buyers will also be able to request a species identification test to confirm that offspring are non-hybrids. Such SNP-based tests are used in aquaculture [86,87] and to distinguish between wolfs and domestic dogs [88]. 

Studies into quantitative traits such as behavioural problems, likelihood to be tamed, adaptation to captivity and longevity, could identify quantitative trait loci (QTL) and associated polymorphisms linked to these traits. These traits could be useful in selecting birds that might adapt better to captive breeding or could live longer in captivity. 

Many pet parrot species are susceptible to the viral Psittacine beak and feather disease (PBFD); it is estimated that up to 10–20% of South African breeding parrot stock is lost due to this virus and it is a worldwide problem [18]. The threat of this virus to natural populations was described in the case where the last wild population of the orange bellied parrot (*Neophema chrysogaster*) was infected by this virus and could be extinct [89]. SNPs linked to natural immunity against this virus can be identified and included onto the SNP chip. This approach was followed by [90] where SNPs linked to natural immunity against the bronchitis virus in chickens was identified. 

A parentage or individual identification test (being species- or genus-specific) could benefit breeders and conservationists. The power of a parentage verification test lays in how consistent a non-biological individual is excluded as a parent based on allele frequencies [91]. The minor allele frequency (MAF) of an individual SNP should be higher than 0.3 to be included in a parentage verification panel [62,91]. When developing such a panel, a large sample size across natural and breeding populations must be included. The number of SNPs included in a panel also influences the exclusion power and [91] suggests the inclusion of at least 200 SNPs. In the study by van der Zwan et al. [69], 960 lovebirds from seven different species were tested at 480 SNPs to compile the parentage verification panel of 195 SNPs for *Agapornis* species.

All parrot species cannot to be viewed as exotic pets as they are the third most popular pet species in large parts of the world. Breeding of parrots, making use of molecular tools, must be incorporated into breeding systems to ensure that healthy, genetically diverse birds are continually bred. A complete SNP chip could be valuable to pet parrot owners, parrot breeders, zoos and parrot conservationists. Bird genomes are known to be smaller due to the erosion of repetitive areas, large segmental deletions and gene loss, and display a high degree of nucleotide similarity, gene synteny and chromosome structure [31,92]. The vast number of species within this order makes the development of a single parrot SNP chip challenging, but a genera-specific chip could be feasible. Therefore, a lot of research within genera will have to be undertaken before SNPs across all species can be identified. This will include sequencing, assembly and annotating more parrot genomes; assembly of existing genomes up to chromosome level (most have been assembled only up to scaffold level, making comparison of genomes a challenge); identifying SNPs for parentage verification and sexing and in depth research into the genes and polymorphisms linked to colour variations. This is not an impossible feat, as a SNP chip was successfully developed across 12 *Eucalyptus* tree species [93]. Cross-species amplification of SNP chips were evaluated using the BovineSNP50, OvineSNP50 and EquineSNP50 chips across domestic and related wild species of bovine, ovine and equine species [94]. Analysis using the OvineSNP50 chip across 16 wild taxa related to the domestic sheep showed that species that had a last common ancestor (LCA) of c. 2.4 million years (MYA) or less had a SNP call rate of >98%, whereas species with a LCA of 33.2 MYA had a call rate of 40.8%. Similar results were seen in the bovine and equine species. They concluded that the SNP call rate decreased by 1.5% with each million year divergence between species. Therefore, the success of the chip would depend on how far evolutionary apart the specific species in a genus are. The implementation of such a SNP chip will, however, be invaluable in parrot breeding systems.

The alarming rate at which parrot species are becoming endangered or extinct places even more emphasize on the role breeders can play in protecting the species. By using molecular tools, breeders can breed healthy, genetically diverse birds that could be re-introduced into the wild, should more species become extinct. Therefore the development of these tests are crucial and should take paramount position in avian research.

## Figures and Tables

**Figure 1 genes-12-01097-f001:**
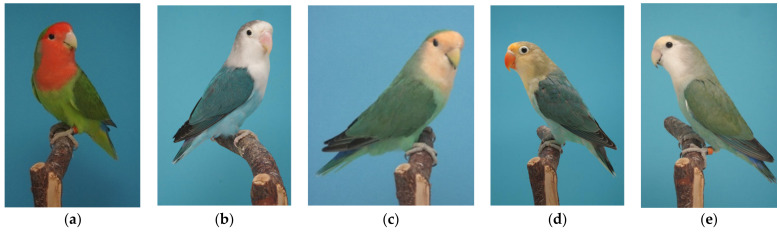
Five *Agapornis* birds: wildtype green *A. roseicollis* (**a**), a blue *A. fischeri* (**b**), an aqua *A. roseicollis* (**c**) and a sapphire *A. fischeri* (**d**) and a turquoise *A. roseicollis* (**e**). All photos curtesy of Mr. Dirk van Den Abeele.

**Figure 2 genes-12-01097-f002:**
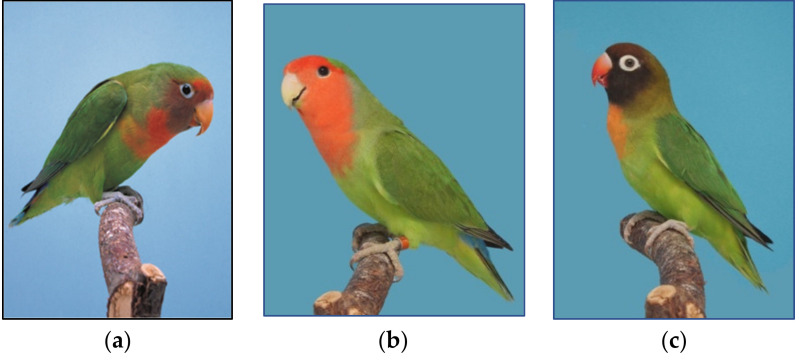
Hybrids of different *Agapornis* species displaying plumage of both species: (**a**) shows a hybrid between *A. roseicollis* and *A. nigriginis*; (**b**) shows a wildtype *A. rosecicollis* and (**c**) a wildtype *A. nigriginis*. All photos curtesy of Mr. Dirk van Den Abeele.

**Table 1 genes-12-01097-t001:** Assembled parrot genomes.

Common Name	Scientific Name	Genome Size (Gb)	Accession Number	Reference
Puerto Rican parrot	*Amazona vittata*	1.58	PRJNA171587	[27]
Scarlet Macaw	*A. macao*	1.11–1.16	PRJNA175470	[28]
Budgerigar	*M. undulatus*	1.2	GigaDB Accession: 1985454	[29]
Peach faced lovebird	*A. roseicollis*	1.1	NDXB01000000	[30]
Kea	*Nestor notabilis*	1.1	PRJNA212900	[31]
Blue fronted Amazon	*Amazona aestival*	1.126	LMAW00000000	[32]
Sun parakeet	*Aratinga solstitialis*	1.16	GCA_902168055.1	[33]
Kakapoo	*Strigops habroptila*	1.148	GCA_004011185.1	[34]
Monk parakeet	*Myiopsitta monachus*	1.168	GCA_017639245.1	[35]

## Data Availability

The study did not report any data.

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
