# Peer review of "Polly Wants a Genome: The Lack of Genetic Testing for Pet Parrot Species"

_genes, 2021, doi:10.3390/genes12071097_

Round 1
Reviewer 1 Report
This review paper by van der Zwan and van der Sluis is timely and informative. The review summarises studies on pet parrot genetic tests, and calls for development of an SNP chip for pet parrots. I do not have major concerns, but have some minor suggestions listed below.
Title, some genetic markers can be developed without genomes - do the authors try to say that genome sequencing is needed to promote genetic testing. Throughout the manuscript, there is limited information on how genomics help develop genetic markers.
L8, the font of “after dogs and cats” seems different, please check.
L26, the citation style looks strange and is not consistent throughout the manuscript
L33, add a reference?
L46, if this is only for USA, then the first sentence in the abstract is not fully accurate.
L63, “the only way” sounds too arbitrary, please consider rephrasing it.
L97, the full genus name should be given if it appears for the first time
L119, the reference [32] should also be given here, because Table 1 does not show 28 species.
Table 1, “Ara macao” should be italicized, and a few parrot genomes are missing:
Sun parakeet, Gelabert et al. 2020, Current biology
Kakapo, Rhie et al. 2021, Nature
Monk parakeet, Huang et al. 2021, bioRxiv
L140, in particular for lovebirds
L145, enters
L146, needs to be revised. What is “one of the countries”, Indonesia or Southern Asia?
L182-184, unclear
L201, MuPKS or MUPKS?
L227, production of
L266, should be “females (W and Z) and one band in males (Z)”
L274-275: I believe the study in [71] is based on the copy number difference of Z-linked genes between males (2 copies) and females (one copy), but not based on SNPs. As the author wrote in line 284 this method is qPCR based which measures the relative copy number. Please correct.
L305-306, the second part of the sentence is speculative, consider removing it.
L316-318, any references?
L323-324, revise this sentence.
L328-329, please note the Z chromosomes are shared by males (ZZ) and females (ZW). Any sexing marker linked to sex should be W-linked which is female-limited, but this is not how sexing markers are developed in birds. Current sexing methods (as the author wrote) are either based on Z-W polymorphism (different intron lengths of CHD1) or Z dosage (qPCR). I doubt if any stable sex-linked SNP (W-linked) can be developed for birds, because the W chromosomes are gene-poor and degenerated, and it is unclear whether there might be a W-linked marker that is conserved across bird species (but see Xu and Zhou 2020 Genes)
L331, what is the cost (economically) to develop such an SNP chip. This need also to be taken into account in breeding practice.
L339, can the author briefly explain why hybrids are not acceptable?
L341, it is unclear how this phenomenon (note the typo) can be reduced after the use an SNP chip.
L350, could also be indels or structural variation, not necessarily SNPs.
L351, the authors were talking about quantitative traits in previous sentences, and it’s a bit sudden to talk about diseases.
L367, repetitive.
Author Response
- Title:
The authors note the reviewer’s concern regarding the title. We have added to Section 3:
The last decade saw the sequencing, assembly and annotation of 31 parrot genomes (see Table 1 for a list of some of the genomes). These reference genomes act as the blueprint from which SNPs, genes and other genetic markers of interest can be identified that could aid in the development of genetic tests for breeders and conservationists. Sequencing and assembling of genomes is not crucial to the development of molecular genetic tests and many markers can be identified without a genome. However, without a draft genome and since many de novo genomes are only assembled up to scaffold level, it complicates the comparison of data from different species. De novo genomes are essential for genotype-to-phenotype association studies, genome comparison studies especially in genome evolution and speciation studies [26]. It is therefore easier to develop cross-species molecular tools from chromosome-level assembled genomes.
- L8, the font of “after dogs and cats” seems different, please check.
Corrected.
- L26, the citation style looks strange and is not consistent throughout the manuscript
Thank you for pointing this out to us. For some reason on the PDF version some of the citations printed in this way, but not in the Word version. All is correct in the new Word version.
- L33, add a reference?
Reference added: Pires, S.F., Olah, G., Nandika, D., Agustina, D. & Heinsohn, R. What drives the illegal parrot trade? Applying a criminological model to market and seizure data in Indonesia. Biological Conservation. 2021, 109098.
- L46, if this is only for USA, then the first sentence in the abstract is not fully accurate.
Abstract amended to “USA”.
- L63, “the only way” sounds too arbitrary, please consider rephrasing it.
Rephrased to: This has been successfully achieved in other pet breeding systems (e.g. dogs and cats) where the development of molecular genetic tools improved breeding systems.
- L97, the full genus name should be given if it appears for the first time
Corrected
- L119, the reference [32] should also be given here, because Table 1 does not show 28 species.
Changed to: The last decade saw the sequencing, assembly and annotation of 31 parrot genomes (see Table 1 for a list of some of the genomes).
(31 = 9 in Table 1 + 22 in reference [36])
- Table 1, “Ara macao” should be italicized
Corrected
- A few parrot genomes are missing:
Sun parakeet, Gelabert et al. 2020, Current biology
Kakapo, Rhie et al. 2021, Nature
Monk parakeet, Huang et al. 2021, bioRxiv
All three were added to Table 1.
- L140, in particular for lovebirds
Corrected
- L145, enters
Corrected
- L146, needs to be revised. What is “one of the countries”, Indonesia or Southern Asia?
Corrected to only Indonesia
- L182-184, unclear
Changed to
Most of the inheritance patterns of known colour variations found amongst lovebirds and budgerigars have been confirmed by test matings as autosomal recessive or sex-linked recessive traits
- L201, MuPKS or MUPKS?
Changed to MuPKS
- L227, production of
Corrected
- L266, should be “females (W and Z) and one band in males (Z)”
Corrected
- L274-275: I believe the study in [71] is based on the copy number difference of Z-linked genes between males (2 copies) and females (one copy), but not based on SNPs. As the author wrote in line 284 this method is qPCR based which measures the relative copy number. Please correct.
Changed to:
A recent study developed a novel qPCR based sexing method by comparing gene copy number variation in a selection of conserved Z-specific genes. Birds from both sexes, from 73 bird species covering 22 orders were tested at nine Z-specific loci (CHRNA6, DDX4, DOCK8, FUT10, LPAR1, PIGG, PSD3, TMEM161B, VPS13A) and three autosomal genes (GGPS1, KIAA1429, MECOM) [75]. These species included seven parrot species (Amazona aestiva (two male, two females), A. amazonica (one male, two females), A. auropalliata (two males, two females), Ara ararauna (one male, one female), budgerigar (two males, two females), Pionites melanocephalus (one male, one female) and Psittacus erithacus (one male, one female)). The Z-specific genes CHRNA6, DDX4, THEM161B and VPS13A were useful in determining the sex of neognath species (the bird clade which include most bird species) and DOCK8, FUT10, PIGG and PSD3 could identify the sex in paleognath birds (the clade which include flightless species). In addition, the LPAR1 gene could identify sex in both clades. It was suggested that the three autosomal genes could be used for normalization and autosomal controls in both clades. As found by Kroczak et al. not all genes amplified in all species. They propose the development of a qPCR molecular sexing method based on these conserved Z-specific genes.
Kroczak et al. [76] compared markers and polymorphisms that have been used routinely and commercially as “universal markers” to identify sex in birds and found that no single universal marker can be applied to sex all birds, especially parrots, successfully. These included the two widely used markers P2P8 and CHD1iA as well as four W/Z length polymorphisms (CHD1iE, CHD1i16, CHD1i9 and NIPBLi16). Their findings suggest that neither the two markers nor the four polymorphisms, when used alone as a single marker, were able to correctly determine the sex of all 135 parrot species. They found CHD1iA, CHD1i0 and NIPBLi16 to perform the most consistent across species. None of these markers could, when used as a single marker, successfully identify sexes of all species tested. They propose the combination of any two of the following four markers: CHD1i16, CHD1i9, NIPBLi16 and CHD1iA, instead of only one universal marker.
- L305-306, the second part of the sentence is speculative, consider removing it.
Removed
- L316-318, any references?
Removed
- L323-324, revise this sentence.
Changed to: When taking the popularity of parrots as pets around the world into consideration as well as the negative effect of inbreeding, the importance of future research on colour polymorphisms, sexing, species identification, health screening and quantitative trait selection cannot be ignored.
- L328-329, please note the Z chromosomes are shared by males (ZZ) and females (ZW). Any sexing marker linked to sex should be W-linked which is female-limited, but this is not how sexing markers are developed in birds. Current sexing methods (as the author wrote) are either based on Z-W polymorphism (different intron lengths of CHD1) or Z dosage (qPCR). I doubt if any stable sex-linked SNP (W-linked) can be developed for birds, because the W chromosomes are gene-poor and degenerated, and it is unclear whether there might be a W-linked marker that is conserved across bird species (but see Xu and Zhou 2020 Genes)
Added:
The polymorphisms tested by Kroczak et al. [76], as a combination of two to four of these polymorphisms can be included on the SNP chip. It should however be noted that the Z chromosome (shared between males and females) are euchromatic and gene-rich, while the W chromosome (only found in females) is gene-poor [84]. Xu & Zhou [84] found that despite the independent evolution of recombination suppression, the female W chromosome in birds still consist of a highly conserved gene content. Further research on polymorphisms located on the W chromosome that can be combined with the polymorphisms suggested by [76] could be combined on a genus specific SNP chip. Autosomal genes, additional to those used by [75], linked to male or female trait expression could be identified in addition to Z-linked markers.
- L331, what is the cost (economically) to develop such an SNP chip. This need also to be taken into account in breeding practice.
Changed to:
Although a genus specific SNP chip could benefit breeders and conservation officials in the fight against illegal poaching and trade, the cost of developing and validating this technology is still relatively high. High costs might make the routine use by breeders impossible.
- L339, can the author briefly explain why hybrids are not acceptable?
Amended and added:
Hybridisation of species as well as genetic introgression is not uncommon in birds [85], especially in a closed colony breeding system where different species of the same genus (e.g. lovebirds) are kept (see Figure 2 for examples of a hybrid bird between two Agapornis species). Breeders also use the crossing of species to introduce colour variations found in one species to another species, or to create a “new” colour variation by combining two colours from two different species [41]. Hybridization could be viewed as either “good” or “bad”, depending on the application. In breeding this could be viewed as “good” since a new species or colour variation could be developed which could boost sales. In conservation this could have negative effects since heterogametic hybrid offspring are often less viable and less fertile than pure birds [85].
- L341, it is unclear how this phenomenon (note the typo) can be reduced after the use an SNP chip.
Typo corrected and added:
Most hybrids are not accepted in breeding societies and at auctions or shows as only pure birds are registered at these bodies [41]. By identifying SNPs linked to specific species and adding these to the SNP chip, it might allow registration bodies to confirm the species or a bird at a show or auction. Buyers will also be able to request a species identification test to confirm that offspring are non-hybrids. Such SNP-based tests are used in aquaculture [86, 87] and to distinguish between wolfs and domestic dogs [88].
- L350, could also be indels or structural variation, not necessarily SNPs.
Changed “SNPs…” “to possible polymorphisms associated with these traits”.
- L351, the authors were talking about quantitative traits in previous sentences, and it’s a bit sudden to talk about diseases.
Information on diseases moved to a new paragraph.
- L367, repetitive.
Removed.
Reviewer 2 Report
The reviewed article discusses the problem of the lack of use of molecular biology tools in the process of breeding parrots as companion animals. The problem described in the manuscript is in my opinion extremely important as many parrot species are endangered and captive individuals may have to be used for reintroduction in the future. Overall the article is written clearly, review made by authors is comprehensive, well presented and discussed. This is the first (or the first I found) article about this subject. I did not find any major problems within text (I did not check English as I am not native speaker) and have only several comments (listed below). In my opinion article needs only minor revision.
Here is the list of small issues I found and would like to point:
- Line 118 – as for now we have 30 parrot genomes in Genbank, please update.
- Line 216 – Parentage and individual identification paragraph – please be aware that tools You are describing (testing parentage) are not so simple as it seems. To be sure that the same allele (in parent and offspring) is the same because of genetic relationship and not by chance is only possible when You will describe allele frequency in whole population. This is very hard (or even impossible in case of parrots) to do.
- Line 254 Sexing – please include new paper about sexing in parrots: “New Bird Sexing Strategy Developed in the Order Psittaciformes Involves Multiple Markers to Avoid Sex Misidentification: Debunked Myth of the Universal DNA Marker”.
- Overall I would also emphasize, that genetic screening of ex-situ parrots population is also very important in context of wild population protection. If we will monitor and breed parrots accordingly we will be able to use them in reintroduction in the future.
Author Response
- Line 118 – as for now we have 30 parrot genomes in Genbank, please
update.
Corrected to 31 genomes (9 genomes in Table 1 plus 22 in reference [36]).
- Line 216 – Parentage and individual identification paragraph – please be aware that tools You are describing (testing parentage) are not so simple as it seems. To be sure that the same allele (in parent and offspring) is the same because of genetic relationship and not by chance is only possible when You will describe allele frequency in whole population. This is very hard (or even impossible in case of parrots) to do.
Added to Section 6: The development of such test will however require extensive research into allele frequencies of populations, both breeding birds and wild populations, across different species and sub-species.
Added to Section 9: A parentage or individual identification test (being species- or genus-specific) could benefit breeders and conservationist. The power of a parentage verification test lays in how consistent a non-biological individual is excluded as a parent based on allele frequencies [90]. The minor allele frequency (MAF) of an individual SNP should be higher than 0.3 to be included in a parentage verification panel [90, 91]. When developing such a panel a large sample size across natural and breeding populations must be included. The number of SNPs included in a panel also influences the exclusion power and [90] suggests the inclusion of at least 200 SNPs. In the study by van der Zwan et al., [69] 960 lovebirds from seven different species were tested at 480 SNPs to compile the parentage verification panel of 195 SNPs for Agapornis species.
- Line 254 Sexing – please include new paper about sexing in parrots: “New
Bird Sexing Strategy Developed in the Order Psittaciformes Involves
Multiple Markers to Avoid Sex Misidentification: Debunked Myth of the
Universal DNA Marker”.
Added to Section 7: Kroczak et al. [76] compared markers and polymorphisms that have been used routinely and commercially as “universal markers” to identify sex in birds and found that no single universal marker can be applied to sex all birds, especially parrots, successfully. These included the two widely used markers P2P8 and CHD1iA as well as four W/Z length polymorphisms (CHD1iE, CHD1i16, CHD1i9 and NIPBLi16). Their findings suggest that neither the two markers nor the four polymorphisms, when used alone as a single marker, were able to correctly determine the sex of all 135 parrot species. They found CHD1iA, CHD1i0 and NIPBLi16 to perform the most consistent across species. None of these markers could, when used as a single marker, successfully identify sexes of all species tested. They propose the combination of any two of the following four markers: CHD1i16, CHD1i9, NIPBLi16 and CHD1iA, instead of only one universal marker.
Added to Section 9:
The polymorphisms tested by Kroczak et al. [76], as a combination of two to four of these polymorphisms can be included on the SNP chip. It should however be noted that the Z chromosome (shared between males and females) are euchromatic and gene-rich, while the W chromosome (only found in females) is gene-poor [84]. Xu & Zhou [84] found that despite the independent evolution of recombination suppression, the female W chromosome in birds still consist of a highly conserved gene content. Further research on polymorphisms located on the W chromosome that can be combined with the polymorphisms suggested by [76] could be combined on a genus specific SNP chip. Autosomal genes, additional to those used by [75], linked to male or female trait expression could be identified in addition to Z-linked markers.
- Overall I would also emphasize, that genetic screening of ex-situ parrots population is also very important in context of wild population protection. If we will monitor and breed parrots accordingly we will be able to use them in reintroduction in the future.
Added to Section 9: The alarming rate at which parrot species are becoming endangered or extinct places even more emphasize on the role breeders can play in protecting the species. By using molecular tools breeders can breed healthy, genetically diverse birds that could be re-introduced into the wild should more species become extinct. Therefore the development of these tests are crucial and should take paramount position in avian research.